# Empirical Research on the Impact of China's Overseas Economic and Trade Cooperation Zones on the Development of Host Countries in the Global Value Chain

**Qing Qin [1] and Churen Sun [2],***

[1]    International Business School, Southwestern University of Finance and Economics, Chengdu 611130, China
[2]    Guangdong International Institute for Strategic Studies, Guangdong University of Foreign Studies,
       Guangzhou 510515, China
*    Correspondence: sunchuren@foxmail.com; Tel.: +86-13331886109

**Abstract:** China's Overseas Economic and Trade Cooperation Zones (COCZs) have become representative platforms of regional cooperation. However, there are few empirical studies exploring their economic and trade effects on host countries. In this study, we comprehensively investigate the impact of COCZs on the host country's GVC participation and positions by constructing a difference-in-difference specification on the industrial level. The main conclusions are: Firstly, COCZs are prominent in promoting the participation in GVCs for host countries while restricting the rise of their positions in GVCs to some extent. The reason lies within the fact that the impact of COCZs on the GVC positions is affected by the host country's factor endowment, innovation level, and business environment. Secondly, for developing countries, the construction of COCZs has a more significant positive effect on their GVC participation. For developed countries, the construction of COCZs can significantly promote their GVC positions. Thirdly, the construction of COCZs has a greater impact on the GVC participation and positions for countries with smaller difference in industrial structure with China, the leading industries in COCZs, and the advantageous industries of host countries.

**Keywords:** Chinese overseas economic and trade cooperation zone; GVC participation; GVC position





## 1. Introduction

Since the 1980s, the rapidly developed value chain economy led by transnational corporations in developed countries has created a new mode of globalization. However, due to the constraints of endogenous and exogenous factors such as geographical disadvantages, the lack of funds, and the limited market size, it is difficult for a large number of enterprises, especially the small- and medium-sized enterprises in developing countries to integrate into the global value chains (GCVs). Even if they participate in GVCs, the weak competitiveness and the injustice in international economic and trade rules caused by the lack of attention to the demands proposed by developing countries would also lead them to the plight of "low-end locking" or the margin areas of the world economy. Moreover, it is undoubted that resolving the global economy contradictions, ensuring the fair development for equal opportunities by promoting the participation, the enhancement of positions in GVCs, and the increase in real gains from trade for developing countries, is a major issue of world development [1].

China has made many attempts to promote the transformation and optimization of industrial and trade structures since the implementation of the Reform and Opening-up Policy, among which the construction of various industrial parks has achieved significant results [2] and has become a typical model for efficient industrial agglomeration and development. The Northwest Suez Economic Zone, which was built by Tianjin Teda Group Co., Ltd. on behalf of Chinese government, is the earliest example of China's construction of industrial parks providing experience for industrial development in developing countries.

In 2005, the Ministry of Commerce of China formally proposed to build overseas economic and trade cooperation zones (COCZs) and began to provide policy and legal support for COCZs. The construction of COCZs began to shift from the stage of spontaneous exploration by enterprises to the stage of government supporting. In 2006, the Ministry of Commerce of China issued the "Basic Requirements and Application Procedures for COCZs" and began to select COCZs to further strengthen the financial and taxation support for them. In 2008, the issue of the "Reply of the State Council on Agreeing to Promote the Construction of COCZs" marked that the construction of COCZs had become a national strategy. In 2010, the Ministry of Commerce of China issued another notice about risk prevention including risk analysis, risk management and insurance to further provide policy support for the construction of COCZs. In 2012, the Ministry of Commerce abandoned the form of examination and approval for the construction of COCZs. Instead, the construction standards of COCZs have been formulated and published. When meeting the standards, the contractor shall apply to the Ministry of Commerce and formally establish the COCZ after the examination. It is required that COCZs must be constructed by newly founded implementing enterprises and complete a series of filling and registration procedures at home and abroad according to relevant policies. Moreover, it is also necessary to complete the infrastructure facilities construction and draw up development plans. The construction of COCZs has entered a rapid development stage.

Since the Belt and Road Initiative was put forward, China has aimed to promote regional cooperation and development in a wider region, a higher level, and a deeper extent among developing countries along the Belt and Road. As the main fulcrum, COCZs have gradually become important platforms for regional industrial capacity cooperation. As shown in Figure 1, until the end of 2018, the number of COCZs had reached 182, they had been distributed in 52 countries, among which are mainly Southeast Asia, South Asia, Central Asia, Russia, and East Africa. Asia is the area where most of COCZs are located, with a total number of 73. Then it is Europe, with a total of 59, mainly concentrated in Russia.

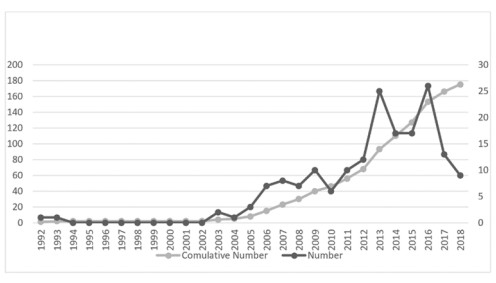

(**a**) The number of COCZs (1992–2018).

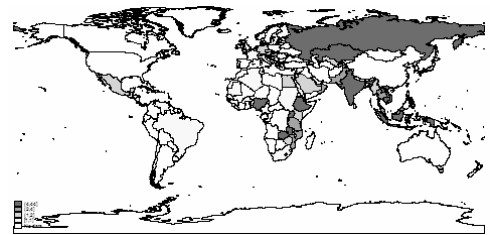

(**b**) The distribution of COCZs.

**Figure 1.** The development trend of COCZs.

With the deepening of the construction of the Belt and Road and the proposal of China participating in the reform of global economic governance system, COCZs are not only significant carriers of China's "Opening-up" Policy, but have also become indispensable grippers to promote the construction and optimization of GVCs, the essence of which is providing chances for the vast number of developing countries to participate in GVCs and enhance their GVC positions, so as to achieve a more fair and reasonable status in the international system of labor division and to earn more gains from trade [3]. As a matter of fact, 70% of COCZs located in the Belt and Road countries, by creating comprehensive industrial parks dominated by the leading industries of the host country with an operation mode integrating R&D, production, processing, transportation, sales, and services, have carried out the initiative of achieving mutual beneficial cooperation. Compared with the traditional bilateral trade and investment agreements, COCZs are more representative in exploiting the advantages of the location and resources of host countries and promoting the formation of sustainable industrial systems [4–6]. On the one hand, COCZs are usually

led by Chinese and host governments or leading enterprises of industries, in the form of "special zones", they provide direct institutional guarantee for enterprises to settle in and to carry out investment and production activities, and help the enterprises to adapt to the complex business environment of host countries and to reduce the costs [7]. On the other hand, most of the COCZs are aligned with the industrial policy orientations of host countries and intend to combine the key areas of China's production capacity cooperation with local comparative advantageous industries, aiming to attract upstream and downstream enterprises to form and extend the industrial chains. For example, the leading industries in the Zambia–China Economic and Trade Cooperation Zone are non-ferrous metal mining, smelting, and processing, and the zone is equipped with sound facilities and services based on Zambia's rich copper ore resources and basic copper product manufacturing industry. COCZs can, to a large extent, promote the formation of a complete non-ferrous metal industry chain and significantly enhance its copper-based industrial-added value and value-added exports [8]. In the long term, COCZs aim at building comprehensive industrial parks that integrate functions such as residence, production, warehousing, transportation, commerce, science research, education, and vacation. They are usually equipped with complete infrastructure, urban service functioning systems, and modern industrial development system. It can greatly improve the investment and business environment of the host countries and form the basis to attract global investment. With the strengthening of specialized production and the expansion of global production networks, local enterprises can obtain better access to participate in global value chains.

It has been proven that COCZs provide a channel for China to export superior production capacity, and significantly promote the bilateral investment and trade between China and the host country [9,10]. Therefore, the establishment of COCZs has created good conditions for enterprises to implement global production layout and to expand international markets. Simultaneously, the impact of COCZs on host countries is also highlighted. The Zambia–China Economic and Trade Cooperation Zone focuses mainly on the development of nonferrous metal industry and the extension of relevant industry chains. It has brought in funds and management experience, enhanced productivity of local workers, and promoted the improvement of trade structure, fully reflecting the development concept of South–South cooperation for mutual benefits [11]. By conducting field interview and a series of comprehensive investigation of COCZs in Africa, it is found that COCZs have introduced a unique, experimental model of industrialization and urbanization for developing countries and have great potential for stimulating local economic and trade growth with the synergy among Chinese government and host governments [8]. Furthermore, it has also been proven that the construction of COCZs has significantly expanded the import and export scale of host countries [12]. Though there is evidence indicating that COCZs are expected to become new engines for the income growth and sustainable development for host countries, most of the existing research adopts the form of case study by selecting representative COCZs [13,14]. The studies investigating their effects are concentrated in fields such as local employment, investment, and traditional trade. A few studies proved that the construction of COCZs have significantly increased the industrial-added value and value-added exports of host countries [4,8], while the lack of systematic theoretical and quantitative analysis leaves room for the further study of their effects on the enhancement of competitiveness and profits in value chain economy. Therefore, combined with the current research of COCZs and the increasingly prominent importance of GVCs, we mainly investigate the effects of COCZs on the GVC participation and positions of host countries.

Our motivation stems from important recent studies that demonstrate the impact of COCZs on host countries and the lack of discussions of their status in GVCs. In this study, based on the EORA data from 2000 to 2015 and the database of COCZs, we comprehensively evaluate the impact of COCZs on host countries' GVC participation and positions by constructing a difference-in-difference (DID) specification on the industrial level. The main conclusion of this study is that the construction of COCZs can effectively promote the GVC participation while has heterogenous effects on GVC positions of host countries, that is, the

construction of COCZs can significantly promote the GVC positions for developed countries, the leading industries of COCZs and the advantageous industries of host countries. The reason lies within the fact that the enhancement of GVC positions is more dependent on the local high-end industrial factor endowments, innovation capability, and the business environment. Industrial cooperation can expand the spillover effects for countries and industries that are with comparative advantages and promote their enhancement of GVC positions. While for those with disadvantages, COCZs brought in the increase in foreign added value in exports in the initial stage and then lead to the deepening of participation degree. When the increase in foreign added value is greater than that of domestic, the GVC position presents a downward trend. Therefore, for most of the host countries as well as developing countries, improving the business environment, including the institutional system and the infrastructure, reducing the production and trade costs, optimizing the factor endowment structure, and improving the innovation ability are the main ways to stimulate the rise of GVC positions.

Compared with previous studies, this study has significant theoretical and practical contributions. From the perspective of theoretical contributions, firstly, it enriches the relevant theoretical research on the global value chain especially on the factors affecting the value chain economy. We comprehensively evaluated the impact of COCZs on GVC participation and position, and expanded the relevant research on the development and optimization of value chain economy by demonstrating the influence of another form of economic and trade cooperation and the mechanisms through which it exerts on GVCs. Secondly, this study enriches the relevant theoretical research on the impact of COCZs. Although scholars have confirmed the positive impact of COCZs on the social and economic development, there is little attention paid to their impact on the GVC status of host countries. We proved the positive effects of COCZs on the value chain economy, we not only helped to measure their effectiveness and adaptability as platforms to promote regional economic and production capacity cooperation, but also helped to deepen the comprehension of their specific roles in promoting the sustainability development of developing countries in GVC and expanded the perspective of relevant research. Thirdly, this paper enriches the research about the correlation of COCZs and GVCs, we also investigated the heterogenous effects and the mechanisms through which the construction of COCZs exert their impact and it helps to reveal the essential characteristic of COCZs. From the perspective of practical contributions, firstly, this study verifies the positive effect of COCZs on the development of host countries in GVCs and provides experiences and references for the construction of regional cooperation platforms to drive the majority of developing countries to achieve the enhancement of competitiveness and profitability and the sustainable development. Secondly, this study provides theoretical and experience support for developing countries and emerging markets to implement industrial cooperation and improve the profitability in the GVCs. The vast number of developing countries have been at disadvantaged stages in GVCs for a long time, and their competitiveness is also relatively weak. The development of developing countries and emerging markets is of great significance for building a sound and sustainable value chain economic system. Our study provides a reference basis for developing countries to formulate industrial cooperation policies and guidelines.

## 2. Theoretical Analysis

### 2.1. The Construction of COCZs and the GVC Participation

The construction of COCZs provides the host countries with more opportunities to deepen the participation degree in GVCs, which is mainly characterized by the inflow of foreign direct investment (FDI) and the expansion of trade scale. On the one hand, COCZs usually set up leading industries based on local resources and factor endowments and advantageous industries. By introducing leading enterprises of relevant industries to absorb more potential investors and enterprises in the upstream or downstream of the industrial chains to settle in, COCZs can attract more foreign direct investment without crowding out domestic investment and generate significant FDI flow effects [10,15]. At

the same time, COCZs can indirectly attract FDI by implementing a series of preferential policies and providing supporting facilities and services [8]. According to the statistics of the Ministry of Commerce of China, by 2019, the COCZs listed in the official website had attracted nearly 5400 enterprises from all over the world to settle in, with a total investment of USD 41 billion. On the other hand, the construction of COCZs can effectively promote the increase in exports and imports scale of host countries, which is mainly due to the efficient and convenient business environment brought by COCZs [12]. Taking the 20 COCZs listed officially by the Ministry of Commerce of China as an example, enterprises settling in can enjoy a relatively large degree of tax preference, the policy period ranges from three to twenty years. Taizhong Rayong Industrial Park provides eight years of corporate income tax exemption for knowledge-based industries, infrastructure construction and high-tech industries. Enterprises in Eastern Industry Zone in Ethiopia are 100% exempt from import customs duties and other taxes related to imports for capital goods such as plants, machinery, and equipment, and the raw materials required to produce export products. Almost all the COCZs provide enterprises with "one-stop" administration and approval services including investment application, registration, customs declaration, issuance of certificate of origin, and some other aspects. Efficient business and customs services support the exports activities of enterprises to a large extent [16,17]. Moreover, for all the listed COCZs, they have finished the construction of most supply systems (water, electricity, roads, mail, communications, heating or steam, and natural gas) by building reservoirs and laying optical fiber cables, natural gas pipelines, sewage, and waste treatment systems. They also provide land, standard industrial workshops, apartments, dormitories, and other rental services. Overall, COCZs provide the enterprises with all necessary facilities and services, and they effectively help enterprises to reduce the costs of production and trade [18] and to overcome the constraints of external conditions. Previous studies have proved that the inflow of FDI significantly promotes the participation in GVCs [19,20], and the changes in the trade scale, especially the supply and demand of intermediate goods, would promote the expansion of value added in exports then the increase in participation degree in GVCs [21,22]. Based on the analysis above, we propose hypothesis 1 as below:

**Hypothesis 1.** *COCZs can promote the increase in the GVC participation degree of host countries by the expansion of foreign direct investment inflow and the exports scale.*

### 2.2. The Construction of COCZs and the GVC Position

COCZs provide the enterprises in host countries with opportunities to participate in global value chains by attracting FDI inflow and expanding the trade scale. However, due to the differences in economic development, the industrial structure, infrastructure, and the institutional environment, the impact of COCZs on the GVC positions varies among different countries and industries.

The resources endowment structure of the host country largely determines the leading industries and the types of COCZs. Therefore, nearly 80% of COCZs located in Asian and African countries are agriculture, light manufacturing, or comprehensive industrial parks. For COCZs located in developing countries, due to the restrictions of economic development and industrial structure, they are more concentrated in building the plants, workshops, and warehouses, upgrading the equipment and strengthening the industrial chains in the initial stage. The substantial increase in capital accumulation and the expansion of exports scale promote the deepening of GVC participation of host countries to a large extent, while its impact on the GVC positions is not very clear. Firstly, COCZs provide a "target orientation" for host countries to attract FDI, but the large amount of FDI may prevent host countries from breaking away from the "trap" of a labor division system, that is, while bringing the capital, technology, and human resources to host countries, it also inhibits the process of innovation and R&D to some extent and solidifies the international labor division status of the host country [23,24]. Secondly, some studies have

shown that industrial policies create a crowding-out effect on other industries, and has a significant inhibitory effect on the GVC positions. Therefore, compared with the participation degree in GVCs, the GVC position which is more dependent on the productivity and value-added creation capability would be affected by the factor endowment, innovation level and business environment of host countries.

From the perspective of factor endowment, the accumulation of advanced industrial factors such as high-end technology [25] and human resources can effectively promote the enhancement of GVC positions [26]. Comparatively, only relying on the number of labor force is not conducive to the rise of GVC positions [27]. From the perspective of innovation capability, the existing research generally believes that the improvement of technology and the total factor productivity are the effective ways to promote the enhancement of GVC positions [28,29]. The innovation of intermediate products in manufacturing industries has a significant promotion effect on the value-added rate of exports, especially on the value-added rate of high-tech manufacturing industries [30]. From the perspective of business environment, studies have shown that the improvement of institutional business environment, such as credit increase and effective government regulations, are beneficial to the production and operation of enterprises [31–33] and exports growth [16,34]. It is believed that the more the days needed to pay taxes, to import or export, or the more the procedures are required to register property, the more it is likely to hinder companies from participating in global value chains, especially in the Middle East and North Africa, the enterprises would be more affected by financing, taxation, and electricity supply [17]. In addition, the improvement of infrastructure such as railways, roads, shipping, and networks promote the rise of a country's industrial output and added value [18] and the expansion of value-added exports [35]. Efficient transportation can provide a comparative advantage to upstream sectors of the value chains [36] and is crucial to the intermediate trade [37].

Therefore, based on the large proportion of COCZs in developing countries and the development situation of local economic and industrial structure, the GVC participation degree of host countries can be promoted by the expansion of FDI and trade scale, but the enhancement of GVC positions is more dependent on host countries' production factor endowment structure, innovation level, and business environment. Moreover, we propose hypothesis 2:

**Hypothesis 2.** *The impact of COCZs on the GVC positions is affected by the host country's factor endowment, innovation level and business environment. For countries with higher concentrating level of advanced production factors, higher innovation level and better business environment, COCZs would have greater effects on their GVC positions.*

*2.3. Heterogenous Effects of Countries and Industries*

Due to the differences in resources, factor endowments, and business environment among countries, the impact of the construction of COCZs on the GVC participation and position of each country and industry are also expected to be different.

From the perspective of countries, firstly, since the enhancement of GVC position is more dependent on the factor endowment, innovation capability, and business environment, compared with the developing countries, COCZs located in developed countries have greater effects on GVC positions of host countries. The reason is that most of the COCZs in developed countries are logistics and high-tech parks, in combination with China's policy orientation of promoting international scientific and technological innovation cooperation and stimulating industrial structure transformation and upgrading through the construction of COCZs, those COCZs mainly conduct industrial R&D activities, aiming at promoting the increase in industrial-added value. In addition, developed countries are usually the places where the advanced production factors are concentrated, such as high-end technology and human capital, etc. The cooperation zones can gather the advantages of each country through industrial cooperation, improve the level of in-

novation, increase the share of domestic value added, and promote the enhancement of GVC positions. Secondly, as one of the few countries in the world with independent and complete industrial systems and all industrial sectors, China plays an important part in the global supply chain, and has the advantages in industrial linkages. The closeness of the production connection between the two countries can not only effectively reduce the costs of carrying out capacity cooperation, but also is more beneficial to expand the positive "spillover effect" [38]. Therefore, the construction of COCZs has a greater effect in promoting the GVC participation and position for host countries with similar industrial structure to China.

From the perspective of industries, the "growth pole" theory can be used to explain the heterogeneous effects of COCZs on the leading and advantageous industries of host countries. The "growth pole" theory was first put forward to explain that the economic growth within a country does not occur in all places in a balanced way. Some regional industries take the lead in development, relying on resources endowments or policy influence, forming growth poles, and then play a spillover and radiation role in the economic development of surrounding areas. This theory has been widely used to formulate development plans for regional economic growth, and COCZs can also be regarded as the growth pole in host counties. On the one hand, the host country provides support for the construction of COCZs from a lot of aspects and the enterprises settled in can take the advantage of policy support to develop. On the other hand, COCZs aim to form the industrial agglomeration by attracting the upstream and downstream enterprises of the relevant industrial chains to settle in to absorb and concentrate the production factors and promote the enhancement of GVC participation and positions. The scale economies and spillover effects also make COCZs growth poles with strong advantages, and drive the development of the surrounding areas and the upstream and downstream enterprises of the industrial chains. Furthermore, the determinant of the leading industries in COCZs is usually based on the comprehensive consideration of the host country's own resources endowment, advantageous industries and China's technological and financial advantages. Therefore, most of the leading industries in COCZs are also local advantageous industries. They are the largest beneficiaries of a series of preferential policies and industrial agglomeration, and the formation of their industrial chains is relatively easy. Taking China–Indonesia Economic and Trade Cooperation Zone as an example, industrial clusters represented by food processing, logistics and warehousing, building materials, and machinery manufacturing industries have been formed. With leading enterprises in each industry settled in, the industrial agglomeration effect is obvious. In Zambia–China Economic and Trade Cooperation Zone, the leading industries in the cooperation zone are non-ferrous metal mining, smelting, and processing, which is based on Zambia's rich copper ore resources and basic copper product manufacturing industry. Therefore, the inflow of production factors and production capacity cooperation formed by COCZs are easier to be gathered in the leading industries of COCZs or the advantageous industries of host countries, thus promoting the economic development of its value chain. Moreover, we propose hypothesis 3 accordingly:

**Hypothesis 3.** *The effects of COCZs on GVC participation and positions of different countries and industries are heterogenous. For developing countries, the construction of COCZs has a more significant positive effect on their GVC participation. For developed countries, the construction of COCZs can significantly promote their GVC positions. The construction of COCZs has a greater impact on countries with smaller difference in industrial structure with China, the leading industries in COCZs and local advantageous industries.*

## 3. Empirical and Data Specification

### 3.1. Empirical Specification

As an economic and trade policy that promotes regional production capacity cooperation, for those countries that have established COCZs, they would be affected by this policy, while the others are not within the scope of the policy. This allows us not only to

observe the difference in GVC participation and the positions of host countries before and after the construction of COCZs, but also to observe the difference between host countries and other countries. The differences in observed samples allow us to design the difference-in-difference (DID) identification strategy to estimate the impact of COCZs on the GVC participation and positions of host countries and to alleviate the endogeneity problem and reduce the estimation errors. Since the time points of the establishment of COCZs varies from countries, we adopted a multi-period DID model to analyze the effects of COCZs [39]. The DID identification strategy is widely used in policy evaluating including the construction of COCZs. Li et al. empirically tested the impact of COCZs on the economic growth of host countries based on the DID identification strategy [40]. Yan et al. found that the establishment of COCZs has significantly expanded the imports and exports of host countries based on DID specification [12]. Similarly, Li et al. exploited the DID strategy to investigate the impact of COCZs on bilateral investment between China and host countries and found a positive correlation between them [10]. Moreover, Li investigated the effect of COCZs on the improvement of infrastructure by exploiting the DID strategy [41]. Therefore, we also adopted the DID specification to investigate the effects of COCZs on the value chain participation degree and positions of host countries separately. Moreover, the estimation specifications are as follows:

$$\ln GVCPat_{ikt} = \alpha_1 + \beta_1 COCZ_{it} + \sum_{m=1}^{M_i} \lambda_1^m Control_{it}^m + \varphi_{ik} + \eta_{kt} + \varepsilon_{1ikt} \tag{1}$$

$$GVCPos_{ikt} = \alpha_2 + \beta_2 COCZ_{it} + \sum_{m=1}^{M_i} \lambda_2^m Control_{it}^m + \delta_{ik} + \phi_{kt} + \varepsilon_{2ikt} \tag{2}$$

Estimation (1) is the specification investigating the effect of COCZs on the GVC participation degree of host countries. $\ln GVCPat_{ikt}$ is the logarithmic value of the participation degree of industry $k$ in country $i$ in year $t$. $\alpha_1$ is the constant term. $COCZ_{it}$ is a dummy variable that takes the value of 1 when the COCZ in country $i$ has been established in year $t$, otherwise it takes the value of 0. $\beta_1$ is the coefficient of the core explanatory variable, if it is significantly positive, it denotes a promotion effect of COCZs on the GVC participation of host countries. If it is significantly negative, it denotes an inhibitory effect on the GVC participation. $Control_{it}^m$ denotes a series of country-level covariates. $\lambda_1^m$ are the coefficients of these covariates and signify the impact of covariates on GVC participation. To further exclude the impact of other unobserved industry-level factors on the results, we add the country–industry fixed effect $\varphi_{ik}$ and the year–industry fixed effect $\eta_{kt}$ in the model. $\varepsilon_{1ikt}$ is the error term of specification (1).

Estimation (2) is the specification investigating the effect of COCZs on the GVC positions of host countries. $GVCPos_{ikt}$ is the indicator of the position in GVCs of industry $k$ in country $i$ in year $t$. $\alpha_2$ is the constant term of specification (2). $COCZ_{it}$ and $Control_{it}^m$ have the same meaning as in specification (1). $\beta_2$ is the coefficient of the core explanatory variable, if it is significantly positive, it denotes a promotion effect of COCZs on the GVC positions of host countries. If it is significantly negative, it denotes an inhibitory effect on the GVC positions. $\lambda_2^m$ are the coefficients of covariates and signify the impact of covariates on GVC positions. We also add the country–industry fixed effect $\delta_{ik}$ and the year–industry fixed effect $\phi_{kt}$. $\varepsilon_{2ikt}$ is the error term of specification (2).

By constructing the DID specifications as above, we shift the perspective from China to the industrial level of host countries and empirically investigate the impact of COCZs on their development in GVCs. It not only helps to improve the comprehension of the specific role of COCZs as regional cooperation platforms, but also provides an empirical analysis framework to comprehensively assess the impact of COCZs on the development of partner countries.

### 3.2. Variables and Data Specification

#### 3.2.1. The GVC Participation and Position Indictors

One of the dependent variables of the regression is the industry-level value chain participation degree of each country. Based on the framework proposed by Koopman et al. [42], the calculation method of GVC participation degree is as follows:

$$GVCPa_{ikt} = \frac{IV_{ikt}}{E_{ikt}} + \frac{FV_{ikt}}{E_{ikt}} \tag{3}$$

where $IV_{ikt}$ and $FV_{ikt}$ represent the domestic added value re-exported to the third countries and the foreign added value of industry $k$ in country $i$ in year $t$, respectively. $E_{ikt}$ is the total exports of industry $k$ in country $i$ in year $t$. The larger the indicator is, the deeper the participation degree is.

The other dependent variable of the regression is the industry-level value chain position of each country. The calculation is also based on the framework proposed by Koopman et al. [42]:

$$GVCPo_{ikt} = \ln(1 + \frac{IV_{ikt}}{E_{ikt}}) - \ln(1 + \frac{FV_{ikt}}{E_{ikt}}) \tag{4}$$

where $IV_{ikt}$, $FV_{ikt}$, and $E_{ikt}$ are identical with the indicators in specification (3), the larger the indicator is, the higher the GVC position is.

We use the latest release of EORA MRIO tables [43] to calculate the GVC participation degree and position indicators. Data from World Input–Output Database are widely used in studies, while the database covers only 41 countries including 27 EU members and 13 major economies around the globe and lacks of information of most developing countries and emerging economies. Compared with that, the EORA MRIO tables have a distinct advantage of offering the largest coverage of countries. The coverage of 189 countries allows us to make investigation on COCZs' impact on developing countries.

#### 3.2.2. The Explanatory Variable

The data of COCZs mainly come from the dataset of Chinese overseas industrial parks (1992–2018) constructed by Li et al. [44]. In addition, we supplemented the information of leading industries and some other aspects of the cooperation zones according to the list of COCZs published by the Ministry of Commerce of China, the official website of COCZs, the official website of the Department of Foreign Investment and Overseas Cooperation of the Ministry of Commerce of China, the special website of COCZs, and the "Going Out Service Platform" of provincial governments. The final sample contains the information of 182 COCZs established from 1993 to 2018, including the name and scale of each COCZ, the country or region where it is located, the domestic implementing enterprise, the construction enterprise, the ownership of the construction enterprise, the category of the park, the leading industries, and other information. The core explanatory variable is determined according to the construction of COCZs. If the cooperation zone entered country $i$ in year $t$, the variable takes the value of 1, otherwise it takes the value of 0.

#### 3.2.3. Covariates

We mainly control several time-varying characteristics that affect the GVC participation and positions of host countries and the probability of a country becoming a host country. We first added the logarithm form of per capital GDP (*lngdp*) and the logarithm form of population (*lnpop*) in regression to control for the impact of the economic scale and population of each country on the production level and then on their GVC status. According to the factor endowment theory, the factor endowment structure determines a country's export structure and the position in value chains to a large extent [45,46]. The abundant natural resources can effectively reduce production costs, enhance innovation capability, and export competitiveness [47]. Therefore, we selected the natural resource output ratio

(*natural*) to measure the natural resource output efficiency. The industrial structure of countries affects their domestic value-added ratio [48], and we added the indictors representing proportion of added value of agriculture and service industries in GDP (*agri, ser*) to control for the impact of industrial structure of host countries. Infrastructure can significantly affect the industrial output, added value in exports of host countries [18]. We added the number of internet servers (*lnnet*) and the number of mobile phones per capital (*lnmoblie*) to measure the effect of infrastructure. In order to control for the factors that may affect the opening level of host countries as COCZs, we further added the indicator variable of free trade zones (*ftz*) in host countries and foreign capital inflow variable (*lnfdi*) obtained from the World Trade Organization website. In addition, studies have shown that a country's institutional environment affects its trade performance. For example, a good legal system is conducive to improving the execution of contracts and thus affecting exports [49–51]. Therefore, we further added the government efficiency index (*gee*), business freedom index (*businessfreedom*), and the trade freedom index (*tradefreedom*) to measure the institutional environment of the host country and to control its impact on the results.

At the same time, since most of the leading industries in COCZs are light industrial manufacturing industries, most of these variables also affect the probability of becoming host countries. In essence, the construction of COCZs is another form of foreign investment, and the natural resources of the host country can strengthen the attraction of investment [52,53], especially for resources-dependent industries. Many studies have also confirmed that enterprises' overseas investment behavior can be affected by the institutional quality of host countries [53,54]. The covariates are mainly from the WDI Database of the World Bank.

After removing the missing observations for key variables, we obtained 37,205 country–industry–year–level observations of 155 countries from 2000 to 2015. Descriptive statistics of these variables are shown in Table 1. The descriptive statistics.

**Table 1.** The descriptive statistics.

| Variable | Obs | Mean | Std. Dev. | Min | Max |
|---|---|---|---|---|---|
| COCZ | 37,205 | 0.132 | 0.338 | 0 | 1 |
| lnpat | 37,205 | −1.077 | 0.943 | −9.407 | 5.765 |
| gvcpos | 37,205 | 1.013 | 0.17 | −1.932 | 1.855 |
| lngdp | 37,205 | 11.412 | 1.74 | 6.37 | 16.748 |
| lnpop | 37,205 | 2.163 | 1.416 | −1.738 | 7.178 |
| natural | 37,205 | 7.497 | 10.747 | 0 | 59.62 |
| agri | 37,205 | 10.964 | 11.201 | 0.031 | 58.652 |
| ftz | 37,205 | 0.064 | 0.245 | 0 | 1 |
| ser | 37,205 | 53.53 | 11.71 | 17.991 | 91.922 |
| lnmobile | 37,205 | 4.235 | 0.859 | −1.568 | 5.47 |
| gee | 37,205 | 0.091 | 0.961 | −2.041 | 2.437 |
| businessfreedom | 37,205 | 66.075 | 15.55 | 20 | 100 |
| tradefreedom | 37,205 | 74.312 | 11.453 | 28.6 | 95 |
| lnnet | 37,205 | 2.978 | 1.365 | −1.862 | 4.587 |
| lnfdi | 37,205 | 21.121 | 2.108 | 12.155 | 27.322 |

Notes: The statistics in this table are calculated from the final sample data.

## 4. Empirical Results and Discussions

### 4.1. Baseline Results

Based on the panel data from 2000 to 2015, we firstly adopted specification (1) and specification (2) to investigate the impact of COCZs on the participation degree and positions of host countries in value chains and the regression results are shown in Table 2. Column (1) shows the estimation results including only the participation degree variable. The coefficient is significantly positive at 1% statistical level, indicating that the construction of COCZs effectively promoted the participation in GVCs for host countries. In order to alleviate the impact of omitted variables on the empirical results, we added a series of

country-level covariates in regression, the estimation results are shown in column (2) and the core coefficient is significantly positive. According to the coefficient value, the GVCs participation degree increased by 12.5% due to COCZs. Columns (3) and (4) show the results of the impact of COCZs on the GVC position. After adding a series of country-level covariates, the coefficient stays negative at 1% statistical level, representing an inhibitory effect of COCZs on GVC position of host countries. The position reduced by 0.6% on average due to COCZs. All the estimating equations include the country–sector and the year–sector fixed effect terms, and the standard errors are clustered at the country–year level.

**Table 2.** Baseline results.

| VARIABLES | (1) lnpat | (2) lnpat | (3) gvcpos | (4) gvcpos |
|---|---|---|---|---|
| COCZ | 0.0990 ** | 0.1251 *** | −0.0043 *** | −0.0063 *** |
|  | (0.041) | (0.041) | (0.002) | (0.001) |
| lngdp |  | 0.3846 ** |  | −0.0163 *** |
|  |  | (0.187) |  | (0.006) |
| lnpop |  | −0.6609 *** |  | 0.0057 |
|  |  | (0.204) |  | (0.008) |
| natural |  | 0.0044 *** |  | 0.0001 |
|  |  | (0.001) |  | (0.000) |
| industry |  | −0.0030 |  | 0.0001 |
|  |  | (0.002) |  | (0.000) |
| fta1 |  | −0.0994 *** |  | 0.0067 *** |
|  |  | (0.024) |  | (0.002) |
| ser |  | 0.0043 ** |  | −0.0002 |
|  |  | (0.002) |  | (0.000) |
| lnmobile |  | 0.0198 |  | 0.0001 |
|  |  | (0.016) |  | (0.001) |
| gee |  | −0.1166 ** |  | 0.0075 *** |
|  |  | (0.053) |  | (0.002) |
| Observations | 52,868 | 37,201 | 52,868 | 37,201 |
| R-squared | 0.946 | 0.948 | 0.973 | 0.985 |
| Year–Sector FE | YES | YES | YES | YES |
| Country–Sector FE | YES | YES | YES | YES |

Notes: ** and *** represent significance at the 5% and 1% levels, respectively. Coefficients of constant terms are not reported in the table.

The estimation results prove that the construction of COCZs has helped the host countries to participate in value chains which lead an effective way to integrate in world economy and would ultimately lead to more gains from trade. This is consistent with previous studies that have verified the positive effect of COCZs on local social and economic development of host countries [12,40,41]. However, they also present a problem that the construction of COCZs may cause a negative effect on the position in value chains for host countries which mainly are developing countries.

The most possible reason lies within the fact that for developing countries, which have relatively weak industrial bases and business environment, the construction of COCZs helps to construct and improve local industrial chains and promote the transformation of industrial structure. Along with the process of industrial agglomeration which includes building plants, updating equipment, and optimizing local resource endowment structure, the local industrial development is characterized by a large amount of foreign capital inflow and the expansion of trade scale [16–19,55–57]. Previous studies have shown that FDI has significantly promoted the participation degree of host countries [19]. Firstly, foreign capital strengthens the value chain integration and connection between the host country and other countries [20,58]. Secondly, FDI alleviates the impact of financial constraints, which show a negative effect on GVC participation [59]. Thirdly, the technology spillovers brought by FDI provide good conditions for technological innovation and value-added creation in host

countries, which effectively promotes the added value in products and services, and then promotes the participation in GVCs [60].

COCZs provide a "target orientation" for host countries to attract FDI, while according to Formula (4), when the increase in the foreign value-added part is greater than that of domestic-added value, the position indicator presents a downward trend. Furthermore, there are two possible reasons for the negative effect of COCZs on GVC positions. Firstly, the large amount of FDI may prevent host countries from breaking away from the "trap" of division of labor, that is, while bringing in the capital, technology, and human resources to host countries, it also inhibits the process of innovation and R&D to some extent and solidifies the GVC status of the host country [23,24]. Secondly, some studies have shown that industrial policies create a crowding-out effect on other industries, and has a significant inhibitory effect on the GVC positions [61]. Therefore, even though there is an overflow effect of FDI, compared with the participation degree in GVCs, the enhancement of GVC position is more dependent on advanced production factors, innovation capability, and institutional environment, which are rare in most host countries. Furthermore, we reckon that there are various effects of COCZs on different countries and industries and we can investigate the heterogeneous effects in Section 5. Though from the point of absolute value, the participation promotion effect of COCZs is far greater than the inhibitory effect on positions, it reveals the dilemma of status rising in value chains and emphasizing the importance of industrial structure upgrading for developing countries.

*4.2. Robustness Checks*

4.2.1. Pre-Existing Trend Test

The baseline regression results reveal a positive effect of COCZs on the participation and a negative effect on the position in GVCs of host countries. However, the potential assumption for utilizing a DID identification strategy is that there is no significant difference between the treatment group and the control group before the policy shock occurs. Corresponding to our study, before the construction of COCZs, the change in the GVC participation and positions of host countries should be consistent with that of the other countries which are without COCZs. In order to verify whether the parallel trend hypothesis holds, we further set up the following models to test the development trend of the participation degree and position in value chains of host countries and other countries before and after the construction of COCZs:

$$\ln GVCpat_{it} = \alpha_1 + \sum \beta_{1t^*} COCZ_{it^*} + \sum_{m=1}^{M_i} \lambda_1 Control_{it}^m + \delta_{1ik} + \phi_{1tk} + \varepsilon_{1ikt} \tag{5}$$

$$GVCpos_{it} = \alpha_2 + \sum \beta_{2t^*} COCZ_{it^*} + \sum_{m=1}^{M_i} \lambda_2 Control_{it}^m + \delta_{2ik} + \phi_{2tk} + \varepsilon_{2ikt} \tag{6}$$

where $COCZ_{it^*}$ takes the value 1 if the COCZ has been in country $i$ in year $t^*$, otherwise it takes the value 0. We selected the year before the construction as the reference year, $\beta_{t^*}$ measures the difference in the participation and position degree between host countries and the other countries. When $t^* < 0$ and $\beta_{t^*}$ is not statistically significant, the parallel trend is satisfied. When $t^* > 0$ and $\beta_{t^*}$ is significant, it measures the effects of COCZs on the participation and position of value chains after entering host countries. We mainly explored the change in participation and position in value chains between host countries and other countries in 3 years before and after the construction of COCZs. Panel (a) and (b) of Figure 2 plot the participation and position variation trend of coefficients within the 95% confidence interval in the time interval of $-4 \le t^* \le 3$, respectively. They present that $\beta_{t^*}$ is not significantly different from zero when $t^* < 0$, indicating that there is no significant difference in the development trend in the four years before the policy implementation between host and other countries and the parallel trend assumption is satisfied. From the perspective of dynamic effects, panel (a) of Figure 2 shows that there is a significant increase in host countries' participation degree since the construction of COCZs. Moreover, panel (b)

presents that the decrease in GVC position appeared one year after the construction of COCZs. Moreover, the changes in the absolute values are consistent with the baseline results that the inhibitory effect of COCZs on GVC position is far smaller than the participation promotion effect.

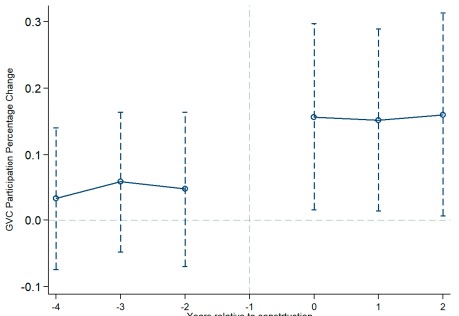

(**a**) Pre-existing trend test of GVC participation.

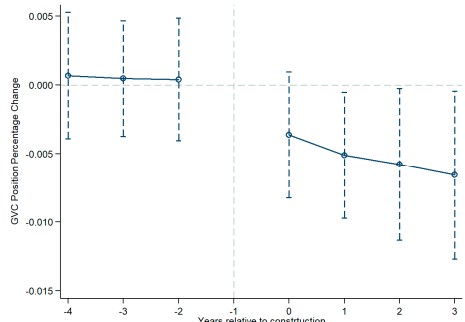

(**b**) Pre-existing trend test of GVC position.

**Figure 2.** Pre-existing trend tests.

### 4.2.2. Placebo Test

In this section, we intend to conduct the placebo test by changing the time points of events to further investigate the impact of COCZs on the participation degree and positions in GVCs of host countries. We added the fictitious time point which is set before the actual construction year in regression, and drop the samples after the actual construction year and to examine whether there is a significant impact of COCZs. As mentioned above, one consumption of using DID specification is that there is no significant difference between countries before the construction of COCZs. Therefore, if the entry time of the COCZ is artificially brought forward, the estimated coefficient of the core variable representing this period is expected to be insignificant. On the contrary, if it is significant, it means that except for the impact of COCZs, there is indeed a systematic difference in the development trend of the GVC participation or position between the treatment group and the control group, and there are some potential unobservable factors other than COCZs that have affected them. In our study, the time points are set as 2 years, 3 years, and 4 years before the actual entry year of COCZs and the dummy variables representing different time points are included separately in regression according to Formulas (1) and (2). The results are shown in Table 3, all the estimated coefficients of the core variable are insignificant, which indicates that the impact of other potential unobservable events can be excluded, and it further confirms the effects brought by COCZs.

**Table 3.** Placebo tests.

|  | (1) | (2) | (3) | (4) | (5) | (6) |
|---|---|---|---|---|---|---|
| VARIABLES | lnpat | gvcpos | lnpat | gvcpos | lnpat | gvcpos |
| placobo_2 | 0.0244 | 0.0001 |  |  |  |  |
|  | (0.019) | (0.002) |  |  |  |  |
| placobo_3 |  |  | 0.0218 | 0.0007 |  |  |
|  |  |  | (0.018) | (0.002) |  |  |
| placobo_4 |  |  |  |  | 0.0254 | 0.0004 |
|  |  |  |  |  | (0.015) | (0.001) |
| Observations | 32,253 | 32,253 | 32,253 | 32,253 | 32,253 | 32,253 |
| R-squared | 0.969 | 0.987 | 0.969 | 0.987 | 0.969 | 0.987 |
| Year–Sector FE | YES | YES | YES | YES | YES | YES |
| Country–Sector FE | YES | YES | YES | YES | YES | YES |

Notes: Coefficients of constant terms are not reported in the table.

### 4.2.3. Other Robustness Checks

Apart from the pre-existing trend and placebo test, we also conducted a series of other robustness checks. Firstly, we replaced the core explanatory variable $COCZ_{it}$ in specification (1) and (2) with the lagged explanatory variable $COCZ_{it-1}$. Considering that there may exist a time lag between the construction of COCZs and when they come into play, the construction of COCZs requires time to build supporting infrastructure, to guide enterprises to settle down, to purchase intermediate products and equipment, and to start production activities, replacing the original explanatory with the lagged dependent variable that matches the actual circumstances and helps to avoid the endogenous problem to a certain extent. The results are shown in columns (1) and (2) of Table 4, and the signs of coefficients of lagged dependent variables remain the same with the baseline results, reconfirming the promotion effect of COCZs on the participation degree and the inhibitory effect on the positions in value chains.

**Table 4.** Robustness tests.

| | **(1)** | **(2)** | **(3)** | **(4)** |
|---|---|---|---|---|
| VARIABLES | lnpat COCZ_lag | gvcpos COCZ_lag | lnpat 2007–2012 | gvcpos 2007–2012 |
| COCZ | 0.1281 *** | −0.0061 *** | 0.2626 ** | −0.0091 ** |
| | (0.041) | (0.001) | (0.131) | (0.004) |
| Observations | 35,733 | 35,733 | 18,950 | 18,950 |
| R-squared | 0.949 | 0.985 | 0.973 | 0.987 |
| Year–Sector FE | YES | YES | YES | YES |
| Country–Sector FE | YES | YES | YES | YES |
| | **(5)** | **(6)** | **(7)** | **(8)** |
| | lnpat 2008–2015 | gvcpos 2008–2015 | lnpat | gvcpos |
| COCZ | 0.1805 ** | −0.0044 * | 0.0953 * | −0.0068 *** |
| | (0.090) | (0.002) | (0.052) | (0.002) |
| Observations | 23,143 | 23,143 | 27,259 | 27,259 |
| R-squared | 0.975 | 0.993 | 0.956 | 0.990 |
| Year–Sector FE | YES | YES | YES | YES |
| Country–Sector FE | YES | YES | YES | YES |

Notes: *, **, and *** represent significance at the 10%, 5%, and 1% levels, respectively. Coefficients of constant terms are not reported in the table.

Secondly, since the construction of COCZs spans a long period of time and this is a long-term policy, in order to avoid its interference with the estimated results, we shortened the policy duration and select two periods of time during which the construction of COCZs is relatively concentrated to perform the empirical investigations. On the one hand, according to Figure 1, the number of COCZs showed a relatively stable growth trend from 2006 to 2012 for Chinese government issued a series of relevant policies to encourage domestic enterprises to establish COCZs in 2005. In 2006, the construction of COCZs has entered a concentrated stage, and it continued to 2012. Therefore, based on the growth trend and the existence of time lag, we first chose the period from 2007 to 2012 to explore the impact of COCZs on the host countries' participation and positions in value chains. On the other hand, considering that the impact of global financial crisis in 2008 may disturb the estimation results, we selected the sample after 2008 to perform the robustness check. Columns (3) and (4) of Table 4. Robustness tests, are the regression results of the period from 2007 to 2012, and columns (5) and (6) are the results of the period after 2008. The results are also consistent with the baseline results that the construction of COCZs has helped the host countries to participate in value chains but somehow inhibit the enhancement of their GVC positions.

Thirdly, we mainly investigated the effects on GVC participation degree and positions brought by COCZs, but there are also some factors highly related to the construction of COCZs, such as signing bilateral free trade agreements (FTAs), bilateral investment agreements (BITs), and the bilateral political relations would affect the GVC participation degree and positions as well. Therefore, in order to reduce the estimation bias of the regression results, we added the indicator representing the bilateral political relations between China and the host country, the indicator variables of signing FTAs and BITs between China and the host country in the benchmark regression. In addition, we further added relevant variables reflecting the stability of the monetary and financial system of the host country, including government expenditure, the ration of broad currency to GDP and domestic wage level. We refer to Bailey et al. [62] to identify bilateral political relations by using the ideal point distance of political inclination between countries measured by the UN General Assembly voting database. FTA data are drawn from the WTO regional trade agreement database and bilateral investment agreement data are from the BIT database of the United Nations Conference on trade and development. The other variables are mainly from WDI database of World Bank. The regression results are shown in columns (7) and (8) of Table 4, after adding these variables, the coefficients of core explanatory variables are statistically significant, indicating that the benchmark estimation results are robust.

## 5. Heterogeneity Analysis

Due to the differences of endowment structures and economic development levels, the impact of COCZs also varies among countries and industries. In this section we further analyze the heterogeneous effects of COCZs on GVC participation and positions among different countries and industries.

### 5.1. Heterogeneity Analysis in Country-Level

Firstly, we investigated the heterogeneous effects of COCZs among developing and developed countries. As is shown in Figure 3, host countries of COCZs include both developed and developing countries. According to the theoretical analysis, the construction of COCZs may lead different ways to develop in value chain economy for different countries. On the one hand, for developing countries, the main aim is to promote industrial agglomeration and to extend the local advantageous industrial chains. Moreover, this form of cooperation leads more to the increase in foreign-added value in exports than that of domestic-added value in the initial stage for COCZs expanded the direct investment inflow and trade scale [20,58]. On the other hand, COCZs located in developed countries which are mainly commercial logistics and advanced processing manufacturing industrial zones provide more platforms for enterprises to conduct high-end technology R&D cooperation for the increase in added value in products and services that promote the enhancement of value chain positions to a larger extent.

Therefore, in this section, we explore the heterogeneity effects of COCZs on countries with different economic development levels. Whether it is a member country of OECD is the standard of dividing developed and developing countries. Columns (1) to (2) in Table 5 are the results of how COCZs affect the value chain participation degree of developed and developing countries, respectively. It shows that COCZs have significant inhibitory effects on the participation degree of developed countries while they can effectively increase the participation degree for developing countries. Columns (3) to (4) are the regression results of how COCZs affect the GVC positions of different countries. It presents that compared with the position inhibitory effect for developing countries, COCZs have a significant promotion effect on developed countries' GVC positions.

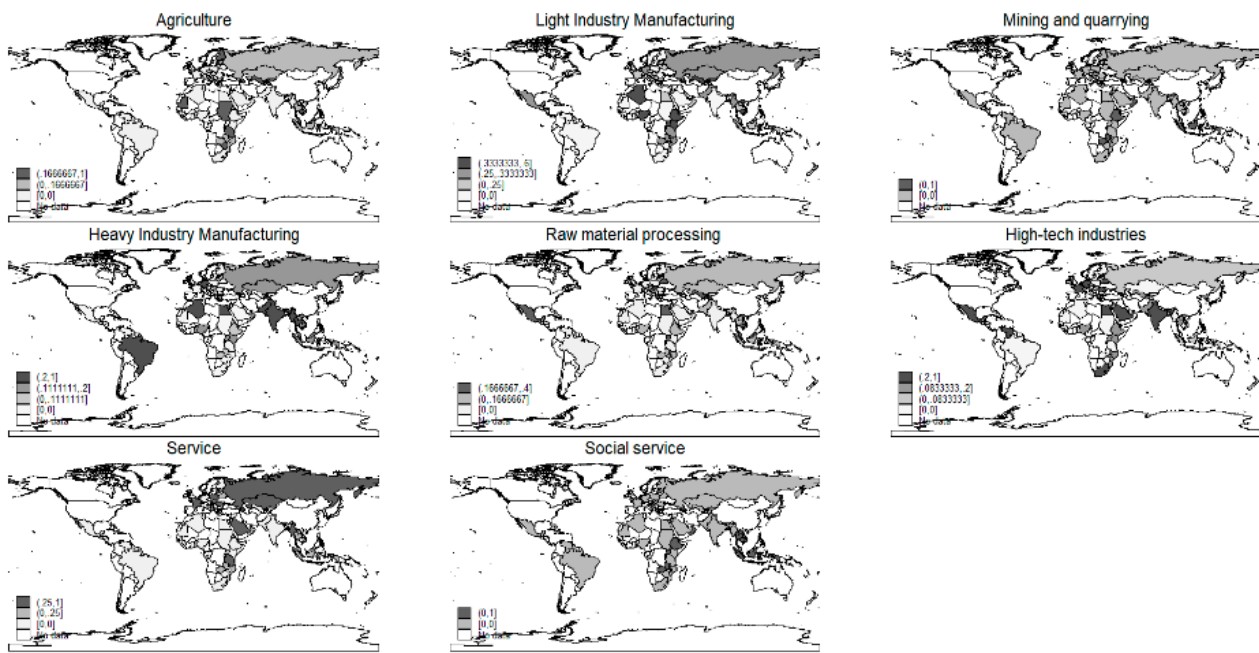

**Figure 3.** The distribution of industries in COCZs around the globe.

**Table 5.** Heterogeneity analysis: developed and developing countries.

| VARIABLES | (1) | (2) | (3) | (4) | (5) | (6) |
|---|---|---|---|---|---|---|
| | lnpat | lnpat | gvcpos | gvcpos | lnpat | gvcpos |
| | Developed Countries | Developing Countries | Developed Countries | Developing Countries | | |
| COCZ | −0.0189 ** | 0.1670 *** | 0.0025 ** | −0.0078 *** | 0.4643 *** | −0.0061 *** |
| | (0.009) | (0.052) | (0.001) | (0.002) | (0.039) | (0.001) |
| Structure × COCZ | | | | | 0.0335 *** | 0.0011 *** |
| | | | | | (0.008) | (0.000) |
| Observations | 8808 | 27,951 | 8808 | 27,951 | 32,006 | 32,006 |
| R-squared | 0.974 | 0.939 | 0.975 | 0.988 | 0.945 | 0.984 |
| Year–Sector FE | YES | YES | YES | YES | YES | YES |
| Country–Sector FE | YES | YES | YES | YES | YES | YES |

Notes: ** and *** represent significance at the 5% and 1% levels, respectively. Coefficients of constant terms are not reported in the table.

This reveals the fact that the industrial cooperation of COCZs is oriented towards specific countries, for developed countries that already play a major role in GVCs with high level of tech and economic development, the industrial cooperation aims more at mutual exchange of information, conducting innovations and the application of more efficient tech or management systems. It is generally believed that the improvement of technology and productivity is one of the effective ways to promote the enhancement of GVC positions [28]. For most developing countries, COCZs mainly help to consolidate the local industrial chains by gathering relevant enterprises, offering essential facilities and to improve business environment. The large amount of FDI inflow helps to promote the GVC participation through the strengthening of value chain connections [58] and it is proven that this effect is more significant among countries along the Belt and Road [3].

Secondly, as an important part of the global supply chain, the industrial connection between the host country and China can not only effectively reduce the costs of carrying out capacity cooperation, but also is more beneficial to the "spillover effect" of COCZs [38]. Therefore, the smaller the difference in the industrial structure between the host country and China is, the higher the probability of carrying out and deepening industrial cooperation

is, thus promoting the GVC participation and positions. We use the ration of added value of the primary, secondary, and tertiary industries in GDP of China and the host country to construct the index of the industrial structure difference between two countries, the formula is as shown below:

$$StructureDiff_{cit} = \left(\sum_{n=1}^{N} |Value_{cnt} - Value_{int}|\right)/3 \tag{7}$$

where $StructureDiff_{cit}$ is the industrial structure difference index, and $Value_{cnt}$ and $Value_{int}$ represent the proportion of added value of industry $n$ in GDP of China and the host country, respectively. The difference index is the average value of the three differences. Columns (5) and (6) of Table 5 are the regression results when adding the interaction terms of the core explanatory variable with the structure difference index. The coefficients of the interaction terms are both negative, suggesting that the smaller the industrial structure difference is, the larger the promotion effect is on the deepening of GVC participation and the enhancement of GVC positions. Since COCZs intend to help local enterprises to improve the productivity and extend local advantageous industrial chains by combining the advantageous production capacity of China and local advantageous industries and most of host countries are weak in industrial foundations [63,64], FDI would have a positive effect on value-added exports of host countries that are similar with China in industrial structure through the convenient industrial cooperation [38].

### 5.2. Heterogeneity Analysis in Industry Level

Firstly, we examined the impact of the construction of COCZs on the GVC participation and positions for the leading industries of COCZs. The main factor determining the leading industry of cooperation zones is the local industrial structure and their advantageous industries. As can been seen in Figure 3, most of the COCZs in developing countries are dominated by light industrial or processing manufacturing such as agriculture, food processing, textiles, and furniture manufacturing industries, and those in European countries are high-tech or logistic and commercial zones. Since COCZs have promotion effects on the GVC participation of developing countries and the GVC positions of developed countries, we further traced the mechanism by investigating the heterogeneous effects on different industries. We divided the industries into agriculture, manufacturing, and service industries and according to the distribution characteristics of COCZs and the different oriented cooperation modes, we regard the former two as dominate industries of COCZs in developing countries and the service industry as the leading industries of developed countries. Columns (1) and (2) in Table 6 show the impact of COCZs on the GVC participation degree of the primary, secondary industries and the service industry for developing countries, respectively. The coefficients of the core explanatory variable are both positive, but different, values, representing that COCZs have greater promotion effects on agriculture and manufacturing industries in developing countries. Comparatively, columns (3) and (4) show the heterogeneous impact of COCZs on the GVC positions of the primary and secondary industries, and the service industry for developed countries, respectively. The significantly positive coefficient in column (3) and the insignificant coefficient in column (4) show that COCZs are beneficial to the enhancement of GVC positions for high value-added industries of developed countries while have no significant effects on the other industries. According to the "growth pole" theory, the leading industries are the key cooperation areas and main beneficiaries of a series of preferential policies and measures, and for developing countries, they more focus on attracting foreign investment, upgrading the production equipment, improving the business environment and the expansion of trade scale so the impact on agriculture and manufacturing industries would be greater. Developed countries are committed to technological research and innovation activities and the high value-added industries are more affected; therefore, the construction of COCZs have conducted specific ways to deepen cooperation for different industries according to local development status.

**Table 6.** Heterogeneity analysis: high and low value-added industries.

| | (1) | (2) | (3) | (4) | (5) | (6) |
|---|---|---|---|---|---|---|
| VARIABLES | lnpat | lnpat | gvcpos | gvcpos | lnpat | gvcpos |
| | Non-OECD High-VA | Non-OECD Low-VA | OECD High-VA | OECD Low-VA | | |
| COCZ | 0.1617 *** | 0.1698 *** | 0.0038 * | 0.0018 | 0.1183 *** | −0.0073 *** |
| | (0.051) | (0.052) | (0.002) | (0.002) | (0.040) | (0.002) |
| RCA × COCZ | | | | | 0.1268 *** | 0.0151 *** |
| | | | | | (0.029) | (0.004) |
| Observations | 9770 | 18,181 | 3060 | 5748 | 37,201 | 37,201 |
| R-squared | 0.944 | 0.929 | 0.951 | 0.992 | 0.949 | 0.985 |
| Year–Sector FE | YES | YES | YES | YES | YES | YES |
| Country–Sector FE | YES | YES | YES | YES | YES | YES |

Notes: * and *** represent significance at the 10% and 1% levels, respectively. Coefficients of constant terms are not reported in the table.

Secondly, in order to explore the heterogeneity effects of COCZs on industries with different levels of competitiveness and to verify whether COCZs are beneficial to the development of local advantageous industries and the improvement of their export competitiveness, we examined the effects of COCZs on different industries with various competitiveness. The revealed comparative advantage index can be used to measure the international competitiveness of a country's products or industries [65]. According to the EORA input–output table, we calculated the industry-level explicit comparative advantage index of each country. The calculation method is shown in Formula (8):

$$RCA_{ij} = \frac{X_{ij}/X_i}{X_{wj}/X_w} \tag{8}$$

where $X_{ij}$ represents the export value of industry $j$ in country $i$, $X_i$ is the total export value of country $i$, $X_{wj}$ indicates the total export value of industry $j$ around the globe, and $X_w$ represents the world's total exports. If the RCA index is less than 1, it means that the proportion of the industry in the total exports of the country (region) is less than the proportion of the industry in the total exports of the world, indicating that the industry of the country (region) is relatively less competitive in the international market. Conversely, when the RCA index is greater than 1, the industry has a comparative advantage or a higher level of competitiveness [65]. We constructed a binary indicator RCA indicating comparative industries when taking the value of 1. The trade data come from UN COMTRADE Database. The core explanatory variable in model (5) and (6) in Table 6 is the interaction term of the RCA indicator and the original core explanatory variable, and the results show that the coefficient of the interaction terms are significantly positive, which indicate that the construction of COCZs flexibly applied the industrial policy to develop local advantageous industries and promote their GVC participation and positions. According to the "growth pole" theory, the reason why the position of value chains of the advantageous industries has been significantly improved compared with the overall industries of host countries is that the advantageous industries are the direct beneficiaries of the preferential policies, such as the tax-free measures implemented by the Taizhong Rayong Industrial Park for knowledge-based industries infrastructure and high-tech enterprises. As the key areas of cooperation, advantageous industries can provide priority to gathering more advanced production factors, realizing the optimization and upgrading of industrial structure and improve the added value of products and services, and take the lead in the status improving in value chains.

## 6. Further Analysis: The Determinants of GVC Positions

The previous analysis shows that, on average, the construction of COCZs is beneficial to the deepening of GVC participation of host countries, but to some extent inhibits the enhancement of GVC positions. However, the heterogeneity effect results show that the construction of COCZs mainly promote the GVC participation of developing countries and the enhancement of GVC positions of developed countries. Simultaneously, COCZs are effective in promoting the enhancement of GVC participation and positions for leading industries and the advantageous industries. This conclusion reveals that the impact of COCZs on the value chain economy depends to a large extent on the economic and industrial development of host countries. This section aims to further explore the factors that affect the impact of COCZs on GVC positions of host countries.

### 6.1. Factor Endowment Structure of Host Countries

The results of heterogenous effects analysis indicate that the GVC positions of developed countries, the leading industries, and the advantageous industries have improved since the construction of COCZs. According to the theoretical analysis, these countries and industries have rich endowments of advanced production factors which are beneficial to the value-added of products and services and then to the enhancement of GVC positions [25–27]. In order to verify this conjecture, we constructed a human resource indicator (HC) of host countries according to the Penn World Table Database. When it takes the value of 1, it indicates that the country is abundant in human resources. We further conducted the regression including the interaction term of the HC index and the original core explanatory variable, and the results are shown in columns (1) and (2) of Table 7. The results show that compared with the deepening of GVC participation, the construction of COCZs has greater effects on the GVC positions for countries with higher level of human resources. In order to compare the effects of different types of production factors, we further constructed a labor indicator (Labor) and add the interaction term to regress. The results are shown in columns (3) and (4) of Table 7. It presents that for host countries with abundant labor force, the construction of COCZs can significantly promote their GVC participation, but the number of labor force has no significant impact on the GVC positions, and this is consistent with previous studies. It is found that relying only on the increase in labor force is not conducive to the enhancement of GVC positions while the improvement of the quality of labor force has a significant positive effect on GVC positions [27]. Specifically, the optimization of human resources structure has significantly promoted the enhancement of the GVC position of China [66]. Li believes that the comparative advantage of labor force provides a chance for developing countries to participate in GVCs, but it is more necessary to adapt the human resources with it to keep the enhancement of GVC positions and the sustainable development [67]. This also reveals that the construction of most cooperation zones is in the early stage of building the industrial chains.

**Table 7.** Moderating effects: factor endowment structure.

|  | (1) | (2) | (3) | (4) |
|---|---|---|---|---|
| VARIABLES | lnpat | gvcpos | lnpat | gvcpos |
| COCZ | 0.2261 *** | −0.0123 *** | 0.0334 | −0.0030 |
|  | (2.98) | (−5.06) | (1.02) | (−1.21) |
| HC × COCZ | −0.1796 *** | 0.0107 *** |  |  |
|  | (−2.60) | (4.35) |  |  |
| Labor × COCZ |  |  | 0.1207 * | −0.0042 |
|  |  |  | (1.90) | (−1.44) |
| Observations | 37,201 | 37,201 | 37,201 | 37,201 |
| R-squared | 0.949 | 0.985 | 0.949 | 0.985 |
| Year–Sector FE | YES | YES | YES | YES |
| Country–Sector FE | YES | YES | YES | YES |

Notes: * and *** represent significance at the 10% and 1% levels, respectively. Coefficients of constant terms are not reported in the table.

### 6.2. Innovation Capability

Innovation is the source of value [68]. Countries with strong innovation capability are less effected by the crowding effect of FDI and are beneficial to the rise of GVC positions. We used the difference value of the number of patents as an indicator (Inno) to measure the gap of innovation levels between the host country and China. We conducted the regression including the interaction term of the index and the original core explanatory variable, and the results are shown in column (1) of Table 8. The estimated coefficient of the interaction term is significantly positive, indicating that the construction of COCZs has a greater impact on promoting the GVC positions of countries that have higher level of innovation. Furthermore, it is consistent with previous studies which have proved that innovation is one of the most effective ways to promote the enhancement of GVC positions [28,29].

**Table 8.** Moderating effects tests.

|  | (1) | (2) | (3) |
|---|---|---|---|
| VARIABLES | gvcpos | gvcpos | gvcpos |
| COCZ | −0.0127 *** | −0.0167 *** | 0.0059 ** |
|  | (−2.89) | (−3.83) | (2.12) |
| Inno × COCZ | −0.0013 ** |  |  |
|  | (−1.96) |  |  |
| Infra × COCZ |  | 0.0204 *** |  |
|  |  | (3.26) |  |
| Costs × COCZ |  |  | −0.0231 *** |
|  |  |  | (−3.89) |
| Observations | 29,778 | 31,814 | 34,602 |
| R-squared | 0.984 | 0.987 | 0.986 |
| Year–Sector FE | YES | YES | YES |
| Country–Sector FE | YES | YES | YES |

Notes: ** and *** represent significance at the 5%, and 1% levels, respectively. Coefficients of constant terms are not reported in the table.

### 6.3. Infrastructure

The construction of COCZs is accompanied by the construction of infrastructure. To verify the effect of infrastructure on GVC positions, we selected a series of indicators to construct an infrastructure index including primary energy use ratio, electricity use ratio, rail transportation volume, container transportation volume, landlines per capita, the number of mobile phones per capita, and the number of broadband subscriptions per capita to indicate the level of infrastructure construction. For each year's data, we first ranked the above indicators of each country separately, then added up the ranking indices according to countries to obtain the overall ranking index, and then normalized them and constructed an infrastructure indicator (Infra). Column (2) in Table 8 is the regression results after adding the interaction term and the coefficient of the interaction term is significantly positive, indicating that the higher the infrastructure construction level of the host country is, the greater the promotion effect of COCZs on its GVC position is. It is generally proved that the improvement of infrastructure can effectively promote the enhancement of productivity and then the development in GVCs for enterprises [35,36]. Specifically, the infrastructure construction provided by China in Africa has a significant positive effect on the productivity of local enterprises and their value-added exports [69] and our results consolidate the view that the construction of COCZs promoted the enhancement of GVC positions.

### 6.4. Business Environment

It is believed that the institutional improvement can promote the enterprises to participate in GVCs and integrate into global market and the effects are greater for developing countries [17]. On the one hand, the implementation of preferential policies and the "one-stop" service in COCZs, such as the exemption of import tariffs, the shortening of the time

for enterprise registration and examination and the customs clearance, can effectively reduce the production and export costs of enterprises, and improve the production efficiency and added value of enterprises. On the other hand, as COCZs are mostly distributed in developing countries, this cluster of industry promotes local SMEs to integrate into GVCs. Therefore, we selected a series of indicators representing production and trade costs, and construct the business cost index (Costs) to evaluate the institutional environment. The indicators selected to measure the business costs include the time required for enterprises to complete property registration, plant construction, tax payment, and contract execution. The regression results by introducing the interaction term are shown in column (3) of Table 8 The coefficient is significantly negative, indicating that the lower the business costs is for a country, the greater the promotion effect of COCZs on its GVC position is. The results are also consistent with the conclusions made by previous studies that sound institutional system helps to promote the enhancement of GVC positions [70] and reaffirms the effect of the COCZs on GVC positions of host countries with different industrial structures, that is, countries with similar industrial structures have lower costs of carrying out production and trade cooperation, which is more conducive to the improvement of their GVC positions.

## 7. Conclusions, Policy Suggestions and Research Outlook

### 7.1. Conclusions and Policy Suggestions

Based on the data from 2000 to 2015, our study comprehensively evaluates the impact of the construction of COCZs on the host country's GVC participation and positions by exploiting a DID specification. The main conclusions are: Firstly, the construction of COCZs can effectively promote the GVC participation but inhibits the promotion of the GVC positions of host countries. Secondly, the GVC participation brought by COCZs is mainly reflected in developing countries. For developed countries, compared with the improvement of participation, COCZs have significantly promoted the GVC positions for them. Thirdly, the construction of COCZs has a more significant positive effect on the GVC participation and positions for countries with similar industrial structure to China, the leading Industries of COCZs and the advantageous industries of host countries. Lastly, the impact of COCZs on the GVC positions is affected by the host country's factor endowment, innovation level, and business environment. For countries with a higher concentrating level of advanced production factors, higher innovation level, and better business environment, COCZs would have greater effects on their GVC position.

Under the international situation of major economies strongly leading the global value chains and trade protectionist policies increasing unprecedentedly, COCZs are important platforms for achieving the production capacity, economic and trade cooperation, and promoting the development of relevant countries and regions. This research enriches the existing literature of the economic effects brought by the cooperation zone, and provides corresponding theoretical support and empirical experience for the subsequent policy adjustment and industrial layout of the cooperation zones. We believe that the construction of COCZs can be further adjusted in the following aspects: Firstly, based on the positive impact of the construction of cooperation zones on local economy and society of host countries, the Chinese government can further strengthen the communication and cooperation with local governments and enterprises and try to provide a good business environment for enterprises in the region and even for the whole country. Secondly, we find that for the leading industries of COCZs and the advantageous industries of host countries, the GVC participation and positions enhancement effects brought by the construction of COCZs are stronger. Therefore, the leading industries of COCZs should be determined according to the local factor endowment structure and their current industrial development situations, and China's production advantages should be provided full ability to promote the development of local advantageous industries through the support of capital and technology. Thirdly, considering that most of the host countries belong to the developing countries along the "the Belt and Road" and the construction of COCZs has a certain inhibitory effect on their GVC positions in the initial stage, the construction of COCZs should focus on exploring

and improving the local resource factor endowment structure on the basis of introducing foreign capital inflows, pay attention to the cultivation of their innovation ability, help to improve local business environment, and establish the long-term sustainable development strategic planning to reverse or decrease the negative impact of the construction of cooperation zones on GVC positions in the initial stage and to promote the enhancement of GVC positions in the future.

### 7.2. Research Outlook

This study comprehensively analyzes the impact of COCZs on the development of host countries in GVCs and enriches the relevant research on the economic impact of COCZs and the factors that affect GVCs. However, there exist limitations in our study: Firstly, the investigation of COCZs in this paper is general and the COCZs are not distinguished by size, operation period and development degree. Therefore, there is a lack of further investigation of the heterogenous effects of various COCZs. Moreover, it is possible to carry out further comparative research of different COCZs characterized by more details. Secondly, due to the limitations of the dataset, we only investigated the impact of COCZs on the industry level of host countries, without identifying the enterprises in COCZs and lack of discussions and empirical tests at the micro-level. Therefore, it is practical to identify enterprises settled in and build a more accurate specification to investigate the impact of COCZs on their development in GVCs in the future.

**Author Contributions:** Conceptualization, C.S.; Software, Q.Q.; Data curation, Q.Q.; Writing—original draft, Q.Q.; Writing—review & editing, Q.Q.; Visualization, C.S.; Supervision, C.S. All authors have read and agreed to the published version of the manuscript.

**Funding:** This paper and the related research are financially supported by the major project of National Social Science Fund "Research on the Construction of 'One Belt and One Road' Regional Value Chain and China's Industrial Transformation and Upgrading" (Project No. 18ZDA039).

**Institutional Review Board Statement:** Not applicable.

**Informed Consent Statement:** Not applicable.

**Data Availability Statement:** The data is not available. The data&code.zip file that we uploaded is only for the reviewer to replicate according to his/her comments.

**Conflicts of Interest:** The authors declare no conflict of interest.

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
