# Peer review of "Empirical Research on the Impact of China’s Overseas Economic and Trade Cooperation Zones on the Development of Host Countries in the Global Value Chain"

_sustainability, doi:10.3390/su15064853_

Round 1
Reviewer 1 Report
In this work, the author on the basis of 2000-2015 data, using the DID specification, through the regression analysis of related variables, the comprehensive evaluation of the construction of China overseas economic and trade cooperation zone on the host country of the global value chain participation and status, then move the research object to developed countries and developing countries, also studied the heterogeneity of different industries in different countries. Finally, the empirical results analyze the conclusion and give suggestions for the construction of overseas economic and trade cooperation zone. This article suggests a minor repair. Other questions in the article are as follows:
1. In the first part, we can update China's latest overseas economic and trade cooperation zone policy, only until 2008.
2. The "growth pole" theory in 2.3 needs to be explained.
3. 3.1 Equations (1) and (2) in α,β,φik,ηkt and others coefficient are not explained.
4. The (b) diagram of figure2 in 4.2.1 ends with ".", which is different from the previous format.
5. 4.2.3 The table 4 mentioned, does not appear in the original text.
6. 5.2 Inside (Balassa 1965) is not explained.

Author Response
Dear Reviewer,
We have uploaded our point-by-point response for your comments and suggestions as a Word file named as "Response to Reviewer 1 Comments". Please see the attachment. We do appreciate the suggestions a lot and we would like to thank you for your review again.
Best regards.

Reviewer 2 Report
The theoretical and empirical framework of the study are appropriate. I suggest major revision, and my comments are as below.
1) The theoretical contribution of the paper should better be strengthened. Thus, appropriate modifications should be made in introduction and/or literature review. Alternatively, the contribution of the manuscript can be stressed in the Section 3.1. in terms of the selection of model variables.
2) It should be persuaded why descriptive stats did not include JB, Skewness and Kurtosis. More importantly, it was mentioned that “The difference in observed samples allows us to design the difference-in-difference (DID) identification strategy to estimate the impact of COCZs on the GVC participation and positions of host countries and to alleviate the endogeneity problem of the estimation.”; however, the advantages of DID in this context should be supported by examples from the literature.
3) The model results should be analyzed by comparing them with the scientific literature.
4) The data (or at least the data sources) and the codes should be provided to the referee to replicate the results.
5) Conclusion section should incorporate policy implications, limitations of the study and suggestions for further research.
Author Response
Dear Reviewer,
We have uploaded our point-by-point response for your comments and suggestions as a Word file named as "Response to Reviewer 2 Comments". Please see the attachment. We do appreciate the suggestions a lot and we would like to thank you for your review again.
Best regards.

Round 2
Reviewer 2 Report
The study is quite comprehensive, and the theoretical and empirical framework of the study is appropriate. I suggest the acceptance of the paper in its current form.